# Genetic Architecture and Meta-QTL Identification of Yield Traits in Maize (*Zea mays* L.)

**DOI:** 10.3390/plants14193067

**Published:** 2025-10-04

**Authors:** Xin Li, Xiaoqiang Zhao, Siqi Sun, Meiyue He, Jing Wang, Xinxin Xiang, Yining Niu

**Affiliations:** State Key Laboratory of Aridland Crop Science, Gansu Agricultural University, Lanzhou 730070, China; m18214360633@163.com (X.L.); sunsiqi1215@163.com (S.S.); hemeiyue0513@163.com (M.H.); 15138079530@163.com (J.W.); 18403693151@163.com (X.X.); niuyn@gsau.edu.cn (Y.N.)

**Keywords:** maize, high-yield breeding, yield component, Meta-QTL, candidate genes

## Abstract

Yield components are the most important breeding objectives, directly determining maize high-yield breeding. It is well known that these traits are controlled by a large number of quantitative trait loci (QTL). Therefore, deeply understanding the genetic basis of yield components and identifying key regulatory candidate genes can lay the foundation for maize marker-assisted selection (MAS) breeding. In this study, our aim was to identify the key genomic regions that regulate maize yield component formation through bioinformatic methods. Herein, 554 original QTLs related to 11 yield components, including ear length (EL), hundred-kernel weight (HKW), ear weight (EW), cob weight (CW), ear diameter (ED), cob diameter (CD), kernel row number (KRN), kernel number per row (KNR), kernel length (KL), grain weight per plant (GW), and kernel width (KW) in maize, were collected from the MaizeGDB, national center for biotechnology information (NCBI), and China national knowledge infrastructure (CNKI) databases. The consensus map was then constructed with a total length of 7154.30 cM. Approximately 80.32% of original QTLs were successfully projected on the consensus map, and they were unevenly distributed on the 10 chromosomes (Chr.). Moreover, 44 meta-QTLs (MQTLs) were identified by the meta-analysis. Among them, 39 MQTLs controlled two or more yield components, except for the MQTL4 in Chr. 1, which was associated with HKW; MQTL11 in Chr. 2, which was responsible for EL; MQTL19 in Chr. 3, which was related to KRN; MQTL26 in Chr. 5, which was involved in HKW; and MQTL36 in Chr. 7, which regulated EL. These findings were consistent with the Pearson correlation results, indicating that these traits exhibited co-linked heredity phenomena. Meanwhile, 159 candidate genes were found in all of the above MQTLs intervals, of which, 29 genes encoded E3 ubiquitin protein ligase, which was related with kernel size and weight. Other genes were involved in multiple metabolic processes, including plant hormones signaling transduction, plant growth and development, sucrose–starch synthesis and metabolism, and reproductive growth. Overall, the results will provide reliable genetic resources for high-yield molecular breeding in maize.

## 1. Introduction

Maize (*Zea mays* L.), as the highest yield and most widely used cereal crop in the world, plays a crucial role in ensuring human food security, the sustainable development of animal husbandry, and modern industry. However, with the reduction in cultivated land area, the increase in extreme weather, and the rapid growth of the global population in recent years, maize yield has been seriously endangered [1,2]. Therefore, how to increase the grain yield of maize has become a major problem faced by breeders. The genetic mechanism of maize grain yield is complex as it is a typical quantitative trait controlled by multiple minor genes [3]. In addition, maize grain yield is the result of the combined effect of multiple agronomic traits, mainly determined by yield components such as ear length (EL), hundred-kernel weight (HKW), ear weight (EW), cob weight (CW), ear diameter (ED), cob diameter (CD), kernel row number (KRN), kernel number per row (KNR), kernel length (KL), grain weight per plant (GW), and kernel width (KW) [4,5]. Through continuous research and analysis, it has been found that the heritability of yield components is much higher than that of grain yield, which has a significant effect on increasing grain yield [6]. At present, our understanding of the molecular mechanisms of maize yield components is still rather limited. Therefore, further elaborating on the molecular mechanisms of maize yield composition can help us better understand the intrinsic connections among these yield components, thereby providing key information for the subsequent breeding of new high-yield maize varieties.

In the field of modern crop breeding, by precisely identifying key quantitative trait loci (QTL) and deeply exploring the application potential of dominant genes, the breeding process of high-yield new maize varieties can be effectively accelerated. Over the past two decades, numerous scholars, both domestically and internationally, have conducted genome-wide association studies (GWASs) and QTL analyses focused on maize yield components [7,8], resulting in the identification of a significant number of single-nucleotide polymorphisms (SNPs) and QTLs. Qu et al. [9] used 447 sweet and waxy maize natural population inbred lines to identify 49 linked SNPs and two crucial candidate genes (*Zm00001d044139* and *Zm00001d000707*) regulating for four yield components of HKW, KL, KW, and kernel diameter based on GWAS. Mei et al. [10] used Yi16 and B73 as parents to construct mapping populations containing 236 F_2_ and 216 F_2:3_. Then, based on the method of inclusive composite interval mapping (ICIM), QTL mapping was conducted for seven yield components in the two environments. As a result, a total of 50 related QTLs were identified in the two sets of populations. In both Coimbatore and Vagalai environments, using the recombinant inbred RIL population as the target population, seven yield components (EW, ED, EL, KRN, KNR, HKW, and grain yield) were located. A total of 22 QTLs were identified, among which 4 QTLs were common in all environments [11]. Although numerous QTLs associated with maize yield components have been detected so far, due to the limitations of various factors such as mapping methods, population types, experimental conditions, and genetic marker density, the confidence intervals (CIs) of the identified QTLs show significant dispersion and the mapping results also exhibit obvious discrete characteristics, making it difficult to form a universal and unified conclusion [12,13]. Therefore, in the actual breeding practice of maize molecular marker-assisted selection (MAS), the number of QTLs that can be precisely located and effectively applied is extremely limited.

Meta-QTL (MQTL) analysis, as a research method integrating multi-source original QTL data, can optimize QTL CIs by constructing mathematical models, significantly improving the accuracy and effectiveness of QTL localization and thereby providing a more reliable basis for the analysis and improvement of complex quantitative traits [14,15]. At present, this method has been widely applied in plant breeding and has achieved good results in the QTL integration of various quantitative traits in various cereal crops. For example, yield components in wheat (*Triticum aestivum* L.) [16] and yield components in rice (*Oryza sativa* L.) [17]. In addition, this method has also made some good progress in maize breeding. Guo et al. [18] conducted MQTL analysis on 428 original QTLs collected related to 23 maize root traits, obtaining a total of 53 MQTLs. By comparing them with 7 genes from rice and 36 genes from *Arabidopsis thaliana*, they ultimately identified 45 candidate genes affecting maize root traits within these MQTL intervals. Li et al. [19] collected 511 original QTLs related to grain yield components, flowering time, plant morphology under drought conditions, and drought tolerance index from 27 published studies for MQTL analysis. A total of 83 MQTLs related to maize drought tolerance were identified, among which 20 were determined as core MQTLs. In addition, 583 candidate genes were identified in 20 core MQTL regions and maize–rice homologous genes. In fact, in previous studies, scholars have utilized bioinformatics techniques to integrate original QTL datasets from various sources and identified many MQTLs related to maize yield components [20,21,22,23]. However, in these studies, due to the differences in the number of integrated original QTLs and the traits involved, the potential candidate genes mined in these MQTL regions remain relatively limited and cannot fully reveal the genetic relationships among the various yield components of maize. Therefore, it is still necessary to integrate and analyze these data using MQTL analysis and to explore the candidate loci that regulate the formation of maize yield.

Clarifying the genetic basis among the components of maize yield can provide valuable information for subsequent MAS breeding, thereby promoting the increase in maize yield. Based on the above considerations, this study integrated 554 original QTLs related to 11 maize yield components (EL, HKW, EW, CW, ED, CD, KRN, KNR, KL, GW, and KW), aiming to identify MQTLs with narrow CIs and stable consensuses through MQTL analyses and predict the key candidate genes that regulate the formation of maize yield within their CIs. These findings can deeply reveal the genetic regulatory networks and molecular mechanism of actions of the components of maize yield, providing a key scientific basis for precisely analyzing the genetic basis of yield traits and ultimately promoting the realization of the goal of high-yield maize breeding.

## 2. Results

### 2.1. Information Collection and Distribution of QTLs Related to Maize Yield Components

To identify 11 consistent genomic regions related to maize yield components controlled by multiple genes, we collected 554 original QTLs from 14 independent studies. Among these studies, four different types of experimental populations were included, i.e., 2 doubled haploid (DH) populations, 11 F_2:3_ populations, 1 F_2_ population, and 4 recombinant inbred lines (RIL) populations, with population sizes varying from 162 to 271 individuals. For the construction of the linkage genetic map, only 1 of the 14 studies used two markers, i.e., simple repeat sequence (SSR) and miniature inverted repeat transposable element (MITE), while the other studies only used SSR as a molecular marker. The length of the maps ranged from 1145.4 to 2769.3 cM. In addition, these populations involved a total of 57 experimental environments, and the QTL mapping method was mainly CIM (composite interval mapping) (Table 1).

For the 554 original QTLs collected, they are unevenly distributed on the 10 chromosomes of maize. Among them, out of 554 initial QTLs, the highest number (100) of QTLs were located on chromosome (Chr.) 1, and the lowest number (30) of QTLs were located on chr.8. The distribution of 11 QTLs related to maize yield components on ten chromosomes shows that HKW (111) accounts for the highest proportion, approximately 20.04%, followed by KRN (105) and EL (88), approximately 18.95% and 15.88%, respectively. The proportion of EW and CW is the same, both 48, approximately 8.66%. Meanwhile, the proportion of KW (8) is the lowest, about 1.44% (Figure 1A). In addition, the majority of individual QTLs phenotypic variance explained (PVE) in the range of 2–20% and had logarithm of odds (LOD) scores between 2 and 10. However, the PVE and LOD values of a single QTL reaching over 20% and 10, respectively, account for only 1.26% and 2.71% of the total original QTLs (Figure 1B,C). Therefore, finding new major QTLs with high LOD and PVE values remains an important task.

### 2.2. Construction and QTL Projection of Consensus Map

By using the IBM2 2008 neighbors map as the reference map, we successfully constructed a comprehensive consensus map using the markers for interval mapping. The total length of this consensus map was 7,154.30 cM, containing a total of 1211 markers, with an average interval of 5.90 cM between the markers. The number of markers on individual chromosomes ranged from a minimum of 91 (Chr.1) to a maximum of 164 (Chr.10). The genetic lengths of individual chromosomes ranged from 534.30 cM (Chr.10) to 1141.4 cM (Chr.1). In addition, among the 554 original QTLs, a total of 445 original QTLs were projected onto the consensus map, accounting for 80.32% of the total original QTLs (Appendix A).

### 2.3. Meta-QTL Analysis and the Relationship Among 11 Components of Maize Yield

According to the lowest Akaike information criterion (AIC) values criteria, we conducted MQTL analysis for 554 original QTLs of 11 maize yield components. Ultimately, 44 MQTLs were identified on ten chrs. of maize, including 336 predicted original QTLs. These MQTLs are unevenly distributed across ten chrs. Among them, the number of MQTLs recognized on chr.1 is the largest, with a total of eight. Next are Chr.2 and 3, each containing six MQTLs. Chr.8 has the fewest MQTLs recognized, with only two. The physical distance of a single MQTL varied from 0.09 to 57.46 Mb, and the 95% CI range of MQTL was from 0.30 cM to 46.10 cM. For these MQTLs, the number of original QTLs they contain ranges from 2 to 21. Among them, MQTL16 on Chr.3 is the most stable MQTL, involving 21 original QTLs. Furthermore, 16 to 19 original QTLs were each integrated on the six MQTLs of MQTL6 (Bin1.08), MQTL9 (Bin2.01–2.02), MQTL13 (Bin2.07), MQTL18 (Bin3.04–3.05), MQTL23 (Bin4.08), and MQTL44 (Bin 10.07), indicating that these MQTL regions also have high stability and there may be a close correlation in the formation of maize yields (Table 2).

Meanwhile, the trait distribution of MQTLs shows that two MQTLs are specific to HKW and are located on Chr.1 and 5, two MQTLs are specific to EL and are located on Chr.2 and 7, and one MQTL is specific to KRN and is located on Chr.3. Similarly, on Chr.2, 3, and 4, we also detected three MQTLs that simultaneously control ED, EL, and KRN. Among the remaining 36 MQTLs, they are respectively associated with various combinations of maize yield components, and each MQTL involves two to nine maize yield components. Among them, MQTL16 (umc2258-umc1030, Bin 3.02–3.04) involves the most yield components, totaling nine, i.e., CW, ED, EL, KL, EW, KNR, KRN, HKW, and GW. Therefore, in subsequent breeding, we can focus on this area to achieve targeted improvement of multiple maize yield components. In addition, we conducted Pearson’s correlation analysis on 11 components related to maize yield to reveal the intrinsic connections among them. The results showed that there was a positive correlation among the 11 yield components. Among them, EL has the highest correlation with HKW and KRN, HKW with KNR, EW with CW and GW, CW with GW, and ED with CD. Their correlation coefficient values are all above 0.90. Meanwhile, there is also a relatively high correlation among other components (Figure 2). It can be seen from this that the formation of maize yield is regulated by multiple components. Finally, it is worth noting that Chr.1, 2, and 3 contain multiple MQTLs, which indicates that they play a significant role in controlling components related to maize yield (Table 2).

### 2.4. Identification of Candidate Genes in MQTLs

Based on the physical distances of the 44 identified MQTLs, we conducted candidate gene screening for these regions. The results showed that among these MQTL regions, a total of 43 MQTL regions identified 159 candidate genes related to the formation of maize yield. Among them, only the MQTL26 region failed to identify potential candidate genes, which might be due to the relatively small physical distance of MQTL26, resulting in fewer functional genes being contained. Furthermore, these candidate genes are unevenly distributed among the 43 MQTLs, with the number of candidate genes for each MQTL ranging from 1 to 13. Among the 159 candidate genes identified, we found that 29 (approximately 18.24%) were E3 ubiquitin protein ligase genes, which play a significant role in regulating kernel size and weight. Meanwhile, the remaining potential candidate functional genes are also involved in a variety of different metabolic processes, including signal transduction of plant hormones (auxin [IAA], cytokinin [CTK], brassinosteroids [BR], and gibberellin [GA]), plant growth and development, sucrose–starch synthesis and metabolism, and reproductive growth (Appendix A). In addition, Dong et al. [12] conducted MQTL analysis and candidate gene identification using 765 original QTLs related to maize yield components collected from 56 independent studies, and reached a consensus on 65 related MQTLs and 5203 candidate genes. And among the 23 MQTLs, 25 functional genes related to maize grain traits and the reported candidate genes were detected. It can be seen from this that MQTL analysis can be used to mine candidate genes related to the formation of maize yield. Meanwhile, based on the distribution of these candidate genes on the 10 Chrs., we found that the highest number of candidate genes were identified on Chr.1 and Chr.2, which indicates that these two regions may be important segments controlling the formation of maize yield (Figure 3).

## 3. Discussion

Nowadays, with the increasingly tight supply of cultivated land resources, the expansion space for grain planting area in China in the future is limited. Therefore, the increase in maize production capacity can only be achieved by raising the grain yield per plant. However, maize yield components are highly hereditary quantitative traits, and their expression is controlled by micro-effect polygenes and is prone to interact with the environment [36,37]. Therefore, in-depth research on the genetic bases and molecular mechanisms among the components of maize yield is conducive to increasing yield and laying a solid foundation for the subsequent breeding of new high-yield maize varieties.

At present, through QTL mapping and GWAS analysis, a large number of genetic loci related to maize yield components have been identified, and some genes acting on yield components have been cloned through mutant studies [38,39,40]. For example, Zhang et al. [41] found overexpressed *Zma-mir1690* and *ZmNF-YA13* maize transgenic plants had significantly affected maize kernel size. Wang et al. [42] discovered an *ids1/Ts6* gene responsible for regulating the number of KRNs in maize through localization cloning and association localization. They also demonstrated through experiments on maize mutants 133D and 116I that an increase in the expression level of this gene would lead to an increase in the number of KRNs in maize. Sun et al. [43] discovered that the transcription factor *ZmBES1/BZR1-5* could positively regulate the size of maize kernels. However, although a large amount of information on the genetic control of yield composition traits in maize has been generated, this QTL information has not yet been widely utilized in maize breeding. This is because the validity of QTL mapping results is influenced by multiple factors, such as experimental conditions, the type and size of the population, and the density of genetic markers.

MQTL analysis is a method that can effectively refine the quantity and location of QTLs to determine stable and high-effect QTLs and identify potential functional genes. In this study, we collected 554 original QTLs related to 11 maize yield components from 14 independent studies for MQTL analysis. A total of 44 MQTLs were identified, and these MQTLs included 336 original QTLs. Meanwhile, we found that among these MQTLs, 39 MQTL regions regulate two or more yield-related traits, indicating that these genomic regions are pleiotropic and can provide key clues for our subsequent analysis of the complex genetic mechanisms underlying maize yield formation. Furthermore, it is worth noting that among these 39 MQTLs we have identified 3 MQTLs that simultaneously regulate maize EL, ED, and KRN, i.e., MQTL14, MQTL20, and MQTL25. Therefore, if the improvement of these three traits is involved simultaneously in subsequent research, we can give priority to choosing these three genomic regions for study. Finally, on Chr.1, 2, 3, 5, and 7, we identified five MQTLs specific to EL, HKW, and KRN. Therefore, in subsequent research, we can focus on these chromosomal regions to search for potential functional genes that regulate EL, HKW, and KRN in maize, thereby accelerating the increase in maize yield.

Next, in order to obtain the potential candidate genes that regulate the formation of maize yield, we projected the identified 44 MQTLs onto the physical reference map B73 RefGen_v4. Ultimately, among these MQTLs a total of 159 candidate genes related to maize yield formation were identified within 43 MQTL intervals. Among these identified candidate genes, we found that 29 of them encode E3 ubiquitin protein ligases. In the ubiquitin–proteasome degradation pathway, the size/weight of kernels is regulated by E3 ubiquitin protein ligase [44]. Therefore, we infer that these genes may regulate the development of maize kernels. Plant hormones, as signaling molecules, play a significant role in the process of plant development. In this study, we identified a total of 38 candidate genes related to the signal transduction of four plant hormones, i.e., IAA, CTK, GA, and BR, in 44 MQTL intervals. Among them, there are 26 genes related to IAA signal transduction. Research has found that IAA plays an important role in the development of maize kernels. It affects the development and filling of kernels by participating in cell differentiation, proliferation, and expansion during the process of kernel development [45]. In addition, among the candidate genes related to CTK signal transduction, we also identified a gene, *Zm00001d002989,* encoding cytokinin oxidase 12. Interestingly, this gene was also identified in a previous study by Dong et al. [12], which indicates that the results of this study have high reliability. In MQTL25, we identified a *Zm00001d053617* encoding cytochrome P450 734A1. Research shows that the regulatory effect of the cytochrome P450 gene on plant growth and development mainly stems from its deep involvement in the two core metabolic pathways of plant hormones: biosynthesis and catabolism [46]. In rice, the overexpression of the cytochrome P450 subfamily gene *GW10* helps to form longer and wider rice grain [47]. Pollen is not only the carrier for maize reproduction but also the core link in the formation of yield. In this study, we identified four candidate genes related to pollen formation, i.e., *Zm00001d032734*, *Zm00001d052422*, *Zm00001d045888*, and *Zm00001d025287*. Among them, *Zm00001d045888* and *Zm00001d025287* are pollen-specific protein genes. Li et al. [48] found that in cotton (*Gossypium hirsutum*) the gradual upregulation of pollen-specific SKS-like protein (PSP231), especially in the post-mitosis stage of pollen, can activate the biosynthesis of callose and promote pollen maturation, thereby avoiding the influence of male infertility.

Plant development is inseparable from the cell growth and development. Among the 11 MQTL intervals, we have identified 20 candidate genes regulating growth and development, among which are five encoded cyclin, i.e., *Zm00001d028365*, *Zm00001d0299*, *Zm00001d032128*, *Zm00001d002788*, and *Zm00001d005293*. In plants, the cell division cycle is regulated by cyclin, and the development of maize ear is closely related to cell division [49]. Therefore, we infer that these genes may affect the development of maize ears by regulating the cell cycle. Sugar is the material basis for the development of maize kernels, and the sufficiency and spatio-temporal dynamics of its supply are crucial for the formation of maize yield. In this study, we identify candidate genes related to multiple sugar metabolisms, and in MQTL12 MQTL14, MQTL16, MQTL22, and MQTL24 we identify the five sugar transporters. As carriers of sugar, sugar transporters play a core role in the transport from sugar sources to sugar reservoirs, influencing the accumulation and distribution of sugar within seeds [50]. In addition, in MQTL14 we also discovered a gene, *Zm00001d007100* (*small kernel 1*), related to the development of maize kernels. The study found that mutations in *small kernel 1* would inhibit the development of embryos and endosperm, thereby affecting the formation of maize yield [51].

In this study, through MQTL analysis, we obtained multiple stable MQTL intervals and potential candidate genes that regulate the formation of maize yield, providing valuable information for high-yield breeding of maize. In subsequent research based on the identified candidate genes, we can create mutants through gene editing technology for backcrossing verification in the field, and thereby cultivate maize varieties with high-yield performance.

## 4. Materials and Methods

### 4.1. Literature Retrieval and QTL Data Collection of Maize Yield-Related Traits

In this study, we conducted a comprehensive literature search through three databases: MaizeGDB (http://www.maizegdb.org/; accessed on 15 May 2025), national center for biotechnology information (NCBI; https://www.ncbi.nlm.nih.gov/; accessed on 15 May 2025), and China national knowledge infrastructure (CNKI; https://www.cnki.net/; accessed on 15 May 2025). We used maize, yield components, and QTL as search terms. We collected QTL location studies related to maize yield components over the past 20 years. Subsequently, we integrated the information of these QTL mapping studies, which included the traits involved in the original QTL, Chr. positions, parental names, LOD values, PVE, mapping methods, population types, population sizes, genetic map sizes, marker types, and the marker intervals in which the QTL was located (Table 1). The yield components involved in these collected original QTLs include EL, HKW, EW, CW, ED, CD, KRN, KNR, KL, GW, and KW, totaling 11 traits. Finally, we screened the collected original QTLs and deleted those with a PVE < 2.0% or LOD < 1.5. In addition, CI, as a key parameter of QTL, is necessary when conducting MQTL analysis. If 95% of the CI is lacking in the collected original QTLs, we can calculate it according to the formula proposed by Darvasi and Soller [52], where N is the size of the original mapping population:(1)$$CI=530/\left(N\times PVE\right)$$(2)$$CI=163/\left(N\times PVE\right)$$

Equation (1) was applicable to backcross and F_2_ populations. Equation (2) was applied to RIL populations.

### 4.2. Consensus Map Construction and QTL Projection

The IBM map (Intermated B73 × Mo17) is a high-density genetic linkage map containing both restriction fragment length polymorphism (RFLP) and SSR markers, while the IBM 2008 neighbors map (http://www.maizegdb.org/; accessed on 22 June 2025) is the latest version of this map, containing 20,925 loci [53]. The maize IBM2 2008 neighbors map was used as the reference map in this study. Then we used BioMercator (V4.2) software (https://urgi.versailles.inra.fr/Tools/BioMercator-V4; accessed on 23 June 2025) to project the original QTL onto the reference map, and thereby established the consensus map of QTLs related to maize yield traits. The prediction of QTLs is based on LOD score and PVE by each QTL, CI, and QTL position. For markers without genetic locations, the QTL is projected onto the consensus map using the marker closest to the QTL flank markers in the references. In addition, if there are QTLs that cannot be mapped to the consensus map or whose mapping positions exceed the consensus map, they should be discarded.

### 4.3. Meta-QTL Analysis of Maize Yield Components

After generating the consensus graph and QTL projection, we used the meta-analysis program in BioMercator (V4.2) software to conduct MQTL analysis. Consistent QTLs were calculated from multiple independent experiments near the same locus on the same Chr. Based on the ratio of the maximum likelihood function, the most likely position and CI of the QTL arrangement on the Chr. were given by Gauss’s theorem. When conducting MQTL analysis, we need to base it on the optimal model values, i.e., Akaike information content (AIC), AIC correction (AICc), AIC 3 candidate models (AIC3), Bayesian information criterion (BIC), and average weight of evidence (AWE) were used to determine the number of potential MQTLs on each Chr. in different experiments. Then, we used the QTL models with the lowest values among at least three of the five models to determine the number of MQTLs on each chr. [54,55], and selected the flanking markers for each MQTL from the Maize GDB data (http://maizegdb.org/; accessed on 16 July 2025) [56].

### 4.4. Identification of Candidate Genes in the MQTLs Interval

To identify the candidate genes in the MQTL regions, we can determine the physical positions of the markers on both sides through the MaizeGDB database. Then, based on the physical location of the MQTL using the genome of maize “Zm-B73-REFERENCE-NAM-5.0”, the potential functional genes within each physical interval were retrieved from the “Plants Ensembl” database (https://plants.ensembl.org/zea_mays/Info/Index; accessed on 24 July 2025) [57]. Gene annotations and their functional data were also retrieved from the same database. Finally, candidate genes related to the formation of maize yield were screened out based on gene annotation and function.

## 5. Conclusions

The components of maize yield are complex quantitative traits controlled by numerous QTLs with relatively small effects. The integration and MQTL analysis of QTLs identified in different contexts can provide valuable information for the fine localization of QTLs and the cloning of key genes. In this study, we collected 554 original QTLs related to 11 components of maize yield, including EL, HKW, EW, CW, ED, CD, KRN, KNR, KL, GW, and KW, for MQTL analysis, and identified 44 stable MQTLs Among them, 39 MQTLs involve two or more yield components, which indicates that the formation of maize yield is influenced by the coordinated effect of multiple traits. Then, based on the physical distances of the identified MQTLs, we identified 159 potential candidate genes that regulate the formation of maize yield within 43 MQTL intervals. These key genes are involved in a variety of metabolic processes, including signal transduction of four plant hormones, plant growth and development, sucrose–starch synthesis and metabolism, and reproductive growth. These results indicate that MQTL analysis is conducive to the rapid determination of candidate genes or loci, and the MQTL regions and candidate genes discovered in this study can be used in future breeding research for MAS to increase the grain yield of maize.

## Figures and Tables

**Figure 1 plants-14-03067-f001:**
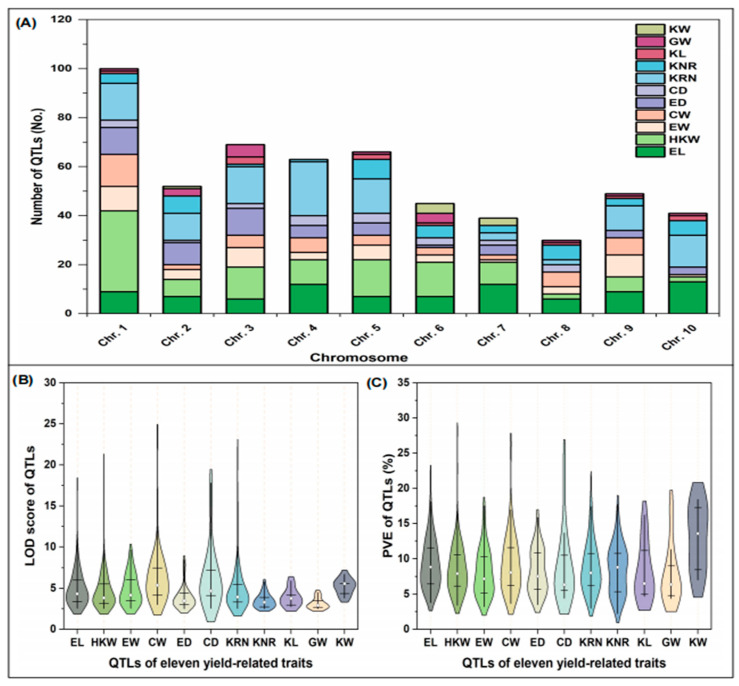
Comprehensive analysis of QTL information related to maize yield components. (**A**) Distribution of QTLs related to maize yield on 10 chromosomes; (**B**) the LOD score values of the QTLs of the 11 yield components; (**C**) the percentage of PVE by each QTL for the 11 yield components.

**Figure 2 plants-14-03067-f002:**
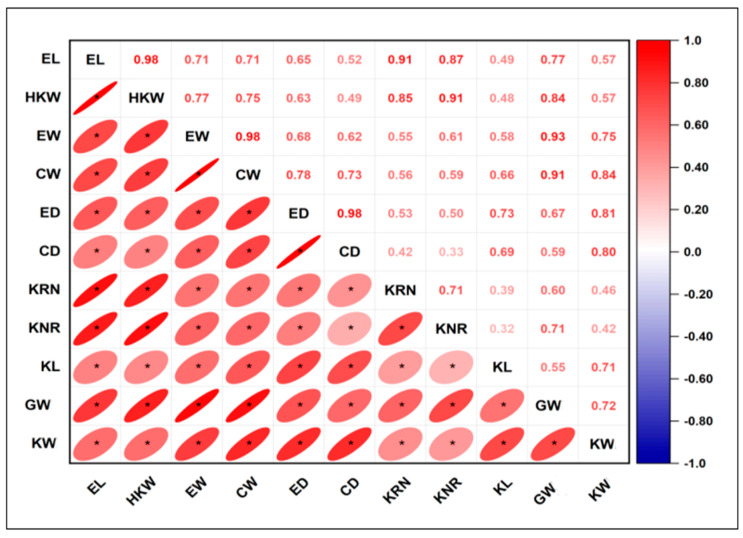
Pearson’s correlation analysis among 11 components of maize yield. * indicates a significant association at the *p* < 0.05 level.

**Figure 3 plants-14-03067-f003:**
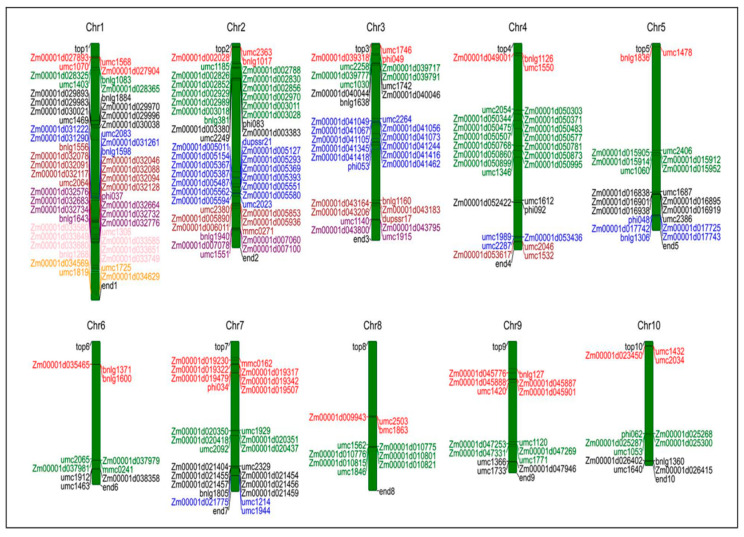
The distribution map of candidate genes identified in 44 MQTL intervals. Different colors on each chromosome represent candidate genes identified within different MQTL intervals.

**Table 1 plants-14-03067-t001:** QTL location information of maize yield components.

Population							QTL Number											
Cross Group	Type	Size	Env.	Marker	Length (cM)	Approach	EL	HKW	EW	CW	ED	CD	KRN	KNR	KL	GW	KW	Reference
HF1 × 11S6169	DH	121	1	200/SSR	1145.4	–	2	1	–	–	–	–	–	–	–	–	–	[1]
Langhuang × TS141	F_2:3_	202	4	213/SSR	1542.5	CIM	11	9	14	13	–	–	–	–	–	–	–	[24]
Chang7–2 × TS141	F_2:3_	218	4	217/SSR	1648.8	CIM	9	8	13	16	–	–	–	–	–	–	–	
BH20 × BH30	F_2:3_	264	1	100/SSR	1281.0	CIM	3	9	2	8	5	7	5	4	6	–	–	[25]
Zheng58 × Chang7–2	F_2:3_	231	2	140/SSR,24/MITE	2245.1	CIM	13	7	5	11	10	15	15	7	–	–	–	[26]
8984 × GY220	F_2:3_	285	3	185/SSR	2111.7	CIM	4	5	–	–	3	–	4	1	–	–	–	[27]
8622 × GY220	F_2:3_	265	3	173/SSR	2298.5	CIM	4	1	–	–	3	–	4	1	–	–	–	
5003 (107) × 178	F_2:3_	210	4	207/SSR	1725.1	CIM	10	12	–	–	11	–	12	11	–	–	–	[28]
Mc × V671	F_2:3_	270	4	256/SSR	1351.7	CIM	–	15	–	–	–	–		–	–	–	–	[29]
Ye478 × D340	F_2:3_	397	7	150/SSR	1478.7	CIM	–	–	–	–	–	–	13	–	–	–	–	[10]
B73 × Yi16	F_2_	236	1	218/SSR	2769.3	ICIM	2	3	–	–	–	–	3	–	–	–	–	[30]
B73 × Yi16	F_2:3_	216	2	218/SSR	2769.3	ICIM	1	4	–	–	–	–	9	–	–	–	–	
02S6140 × KSS22	F_2:3_	158	1	303/SSR	2626.5	–	2	–	–	–		–	–	–	–	–	–	[31]
8984 × GY220	RIL	282	4	216/SSR	2285.3	CIM	9	10	8	–	10	–	14	9	–	5	–	[32]
8622 × GY220	RIL	263	4	208/SSR	2217.2	CIM	7	19	6	–	2	–	10	1	–	5	–	[33]
Zheng58 × Chang7–2	DH	162	4	119/SSR	2315.0	CIM	8	8	–	–	6	–	12	8	–	7	–	
Huang C × Xu 178	RIL	166	4	217/SSR	2438.2	CIM	–	–	–	–		–			–	5	8	[34]
Xu178 × K12	RIL	150	4	191/SSR	2069.1	ICIM	3	–	–	–	2	–	4	2	–		–	[35]

RIL: recombinant inbred line; DH: double haploid; Env.: environment; EL: ear length; HKW: hundred-kernel weight; EW: ear weight; CW: cob weight; ED: ear diameter; CD: cob diameter; KRN: kernel row number; KNR: kernel number per row; KL: kernel length; GW: grain weight per plant; KW: kernel width; CIM: composite interval mapping; ICIM: inclusive composite interval mapping.

**Table 2 plants-14-03067-t002:** Meta-QTL analysis results of 11 maize yield components.

Trait	MQTL	Chr.	Position (cM)	QTLs Number	Bin	Marker Interval	CI (cM)	Physical Interval (Mb)	Contig
CW, EL, HKW, ED, KRN	MQTL1	1	145.30	11	1.02	umc1568–umc1070	141.80–148.80	15.80–17.67	ctg6–ctg9
EW, HKW, KRN, KNR, GW	MQTL2	1	205.90	8	1.02–1.03	bnlg1083–umc1403	198.30–210.60	29.26–33.13	ctg9–ctg10
CW, ED, EL, EW, KL, HKW, KRN, KNR	MQTL3	1	431.20	9	1.05	bnlg1884–umc1469	419.80–438.00	91.88–115.66	ctg23–ctg28
HKW	MQTL4	1	519.70	2	1.05–1.06	umc2083–bnlg1598	506.40–532.80	179.61–187.92	ctg37–ctg38
CD, ED, EL, HKW	MQTL5	1	662.40	10	1.07	bnlg1556–umc2064	658.60–668.30	208.48–220.66	ctg44
CD, CW, ED, EL, EW, HKW, KRN	MQTL6	1	748.10	16	1.08	phi037–bnlg1643	722.40–768.50	228.38–238.45	ctg46–ctg50
CD, KRN	MQTL7	1	874.80	2	1.09	umc1306–bnlg1268	866.00–898.70	265.24–273.4	ctg57
CW, KRN	MQTL8	1	1113.00	4	1.12	umc1725–umc1819	1096.50–1119.20	296.28–298.64	ctg65–ctg66
ED, EL, EW, HKW, GWP, KRN, KNR	MQTL9	2	56.40	17	2.01–2.02	umc2363–bnlg1017	51.10–62.60	4.16–4.9	ctg69–ctg71
CW, EW, KRN	MQTL10	2	229.20	4	2.03–2.04	umc1185–bnlg381	213.40–241.60	21.39–29.89	ctg74
EL	MQTL11	2	287.50	3	2.04	phi083–umc2249	281.60–293.20	41.22–44.26	ctg77–ctg78
ED, HKW, KRN, GW	MQTL12	2	372.10	11	2.05–2.06	dupssr21–umc2023	366.90–380.40	153.48–182.63	ctg90–ctg96
EW, KRN, KW, HKW, GW	MQTL13	2	421.90	16	2.07	umc2380–mmc0271	414.80–435.20	190.73–197.18	ctg98–ctg100
ED, EL, KRN	MQTL14	2	584.30	3	2.08–2.09	bnlg1940–umc1551	574.50–596.00	219.78–223.63	ctg105–ctg108
ED, EL, HKW	MQTL15	3	7.60	3	3.00–3.01	umc1746–phi049	6.70–8.80	1.63–1.73	ctg111
CW, ED, EL, KL, EW, KNR, KRN, HKW, GW	MQTL16	3	141.50	21	3.02–3.04	umc2258–umc1030	127.00–158.20	10.08–14.94	ctg112–ctg114
CW, ED, HKW	MQTL17	3	189.00	5	3.04	umc1742–bnlg1638	188.20–189.90	23.37–26.31	ctg116
C, ED, EL, EW, HKW, GW	MQTL18	3	280.70	19	3.04–3.05	umc2264–phi053	260.30–297.60	90.04–126.49	ctg121–ctg124
KRN	MQTL19	3	508.40	3	3.06	bnlg1160–dupssr17	490.60–517.20	187.49–194.08	ctg138–ctg139
ED, EL, KRN	MQTL20	3	612.60	10	3.08	umc1140–umc1915	608.40–616.70	209.72–210.93	ctg145–ctg146
CD, CW, EL, HKW	MQTL21	4	141.50	6	4.03	bnlg1126–umc1550	135.30–152.00	11.93–14.88	ctg158–ctg159
CD, ED, EL, KRN	MQTL22	4	303.20	5	4.05	umc2054–umc1346	302.50–304.30	78.64–136.1	ctg174
CW, EL, EW, HKW, KRN	MQTL23	4	511.70	17	4.08	umc1612–phi092	493.00–522.10	187.53–190.23	ctg185–ctg187
CW, EW, HKW, KRN	MQTL24	4	608.10	13	4.09	umc1989–umc2287	599.60–618.10	230.28–232.44	ctg198
ED, EL, KRN	MQTL25	4	663.60	5	4.09–4.10	umc2046–umc1532	657.00–671.90	236.3–237.5	ctg201
HKW	MQTL26	5	82.40	2	5.01	umc1478–bnlg1836	79.20–87.60	4.49–4.58	ctg204–ctg205
CD, EL, KNR	MQTL27	5	313.40	6	5.04	umc2406–umc1060	311.30–317.60	127.64–136.49	ctg233–ctg234
ED, KL, HKW, KRN	MQTL28	5	420.50	9	5.05	umc1687–umc2386	411.10–428.00	176.94–180.49	ctg240–ctg242
ED, EL, EW	MQTL29	5	543.80	7	5.07	phi048–bnlg1306	536.60–564.60	204.66–207.03	ctg252
CW, HKW	MQTL30	6	70.10	2	6.01	bnlg1371–bnlg1600	68.70–71.80	27.97–28.31	ctg262
EL, HKW	MQTL31	6	308.00	5	6.05	umc2065–mmc0241	307.10–308.80	142.75–144.32	ctg285
CD, KL, EL, KW	MQTL32	6	386.30	8	6.06	umc1912–umc1463	384.70–389.90	154.35–155.33	ctg287
ED, HKW, KRN	MQTL33	7	164.90	3	7.02	mmc0162–phi034	154.00–170.50	20.34–38.91	ctg304
EL, ED, HKW, KRN	MQTL34	7	248.20	5	7.02	umc1929–umc2092	246.80–250.60	107.95–114.7	ctg310–ctg312
CW, ED, HKW, KRN	MQTL35	7	385.70	5	7.03	umc2329–bnlg1805	382.10–389.80	151.28–153.7	ctg322
EL	MQTL36	7	463.30	2	7.03–7.04	umc1214–umc1944	462.20–464.50	162.29–162.85	ctg322–ctg323
CW, EL	MQTL37	8	239.00	2	8.03	umc2503–bmc1863	238.90–239.20	90.3–91.64	ctg340
EL, EW, KNR, GW	MQTL38	8	343.50	9	8.05	umc1562–umc1846	336.40–351.40	125.43–130.27	ctg354
EL, KRN	MQTL39	9	218.20	3	9.03	bnlg127–umc1420	214.50–225.40	33.59–50.02	ctg375–ctg377
ED, EL, KNR, GW	MQTL40	9	302.00	11	9.04	umc1120–umc1771	298.10–305.30	121.84–127.58	ctg384–ctg385
EL, HKW, KRN	MQTL41	9	479.70	8	9.06	umc1366–umc1733	478.10–480.50	145.6–146.47	ctg389
EL, HKW, KRN, KNR	MQTL42	10	96.30	5	10.02	umc1432–umc2034	91.40–103.50	5.77–6.38	ctg392
EL, KL, HKW	MQTL43	10	249.40	3	10.04	phi062–umc1053	243.90–255.60	111.86–114.31	ctg411–ctg412
ED, EL, EW, KNR, KRN, GW	MQTL44	10	460.10	18	10.07	bnlg1360–umc1640	452.70–467.00	144.91–145.62	ctg419

Chr.: chromosome; CI: confidence interval.

## Data Availability

Data are contained within the article and Appendix A.

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
