# Peer review of "Genetic Architecture and Meta-QTL Identification of Yield Traits in Maize (Zea mays L.)"

_plants, 2025, doi:10.3390/plants14193067_

Round 1
Reviewer 1 Report
Comments and Suggestions for Authors
In the present manuscript, the author addresses an important issue on the genetic basis of yield-related traits in maize. The authors mainly focused on meta-QTL analysis, which contributes to refining QTL regions, facilitating marker-assisted breeding. The study is well-structured, and the experimental design appropriately supports the conclusions drawn. However, the similarity index shows 27%, which needs to be reduced to at least less than 20%. The authors need to provide authentic pages and line numbers in the original manuscript. The following points should be considered:
2.4. Identification of Candidate Genes in MQTLs:
The author mentioned the different metabolic processes in Supplementary Table 1, but they didn’t put any supplementary files.
Figure 2. Consensus maps built on fourteen original map datasets:
The authors need to put the actual chromosome length (Mbp) in the corresponding chromosome numbers.
4.1. Literature Retrieval and QTL Data Collection of Maize Yield-Related Traits:
The author fails to discuss how many plants they used in each type of generation, e.g., doubled haploid (DH) populations, F2:3 populations, F2 populations, recombinant inbred lines (RIL) populations etc. Please explain the details of experimental procedures here.
The author needs to check the reference lists of those that are cited in the text. All the titles and other contents should be unique to the journal format.
Author Response
Dear Editor and Reviewers
Thank you for your letter of – and for the referee’s comments concerning our manuscript, “Genetic Architecture and Meta-QTLs Identication of Yield Traits in Maize (Zea mays L.) (Manuscript ID: plants-3890154)”. We have carefully studied these comments and have made corresponding corrections to the manuscript, which we describe in detail below. We would like to re-submit the manuscript and that for possible publication on the Special Issue: “Recent Advances in the Genetics, Genomics and Breeding of Cereal Crops” of Plants. Thank you very much for your time and consideration.
Editor:
Your manuscript has now been reviewed by experts in the field and can be found with the review reports at: https://susy.mdpi.com/user/manuscripts/resubmit/7ffe290692e2f8382a73f4255b5daa94 Please revise the manuscript found at the above link according to the reviewers' comments and upload the revised file within 5 days.
Thanks for the positive comments of you and all reviewers for our manuscript. As suggested, we have carefully revised and improved our manuscript using the “Track Changes” function of the manuscript at the above link. We then have re-submitted the manuscript within the allotted time.
Thank you for your consideration.
(I) Ensure all references are relevant to the content of the manuscript.
Thanks for the positive comments. As suggested, we have carefully checked all references. We then have re-submitted the manuscript.
Thank you for your consideration.
(II) Highlight any revisions to the manuscript, so editors and reviewers can see any changes made.
Thanks for the positive comments. As suggested, we have carefully revised and improved our manuscript using the “Track Changes” function of the manuscript. We then have re-submitted the manuscript.
Thank you for your consideration.
(III) Provide a cover letter to respond to the reviewers’ comments and explain, point by point, the details of the manuscript revisions.
Thanks for your positive comments for our manuscript. As suggested, we have carefully revised and improved our manuscript. In addition, we have prepared a detailed response letter to all reviewers for each point, and then have re-submitted the manuscript.
Thank you for your consideration.
(IV) If the reviewer(s) recommended references, critically analyze them to ensure that their inclusion would enhance your manuscript. If you believe these references are unnecessary, you should not include them.
Thanks for your positive comments for our manuscript. As suggested, we have carefully checked and revised the References. We then have re-submitted the manuscript.
Thank you for your consideration.
(V) If you found it impossible to address certain comments in the review reports, include an explanation in your appeal.
Thanks for your positive comments for our manuscript. As suggested, we have carefully revised and improved our manuscript. In addition, we have prepared a detailed response letter to all reviewers for each point, and then have re-submitted the manuscript.
Thank you for your consideration.
If your manuscript requires improvement to the language and/or figures, you may consider MDPI Author Services: https://www.mdpi.com/authors/english. Please note the status of this invitation “Publish Author Biography on the webpage of the paper” - https://susy.mdpi.com/user/manuscripts/resubmit/7ffe290692e2f8382a73f4255b5daa94. If you wish to publish your biography, please complete it before your manuscript is accepted.
Thanks for the positive comments. As suggested, we have carefully checked and revised the English language of the manuscript. We then re-submitted the manuscript.
In addition, thanks for your invitation, we decided not to publish our biography.
Thank you for your consideration.
Please do not hesitate to contact us if you have any questions regarding the revision of your manuscript or if you need more time. We look forward to hearing from you soon.
Thanks for your positive comments for our manuscript. As suggested, we have carefully revised and improved the manuscript using the “Track Changes” function of our manuscript at the above link. We then have re-submitted the manuscript within the allotted time.
Thank you for your consideration.
Reviewer 1
Comments and Suggestions for Authors
In the present manuscript, the author addresses an important issue on the genetic basis of yield-related traits in maize. The authors mainly focused on meta-QTL analysis, which contributes to refining QTL regions, facilitating marker-assisted breeding. The study is well-structured, and the experimental design appropriately supports the conclusions drawn. However, the similarity index shows 27%, which needs to be reduced to at least less than 20%. The authors need to provide authentic pages and line numbers in the original manuscript. The following points should be considered:
Thanks for your positive comments. As suggested, we have carefully revised the manuscript to decrease the similarity index, which has reached 2.7% (Fig. 1-1) by the analysis of China National Knowledge Infrastructure (CNKI).
Fig. 1-1 The results of similarity index detection by China National Knowledge Infrastructure.
We then re-submitted the manuscript.
Thank you for your consideration.
2.4. Identification of Candidate Genes in MQTLs:
- The author mentioned the different metabolic processes in Supplementary Table 1, but they didn’t put any supplementary files.
Thanks for your comments. Due to the negligence in the last submission, we were unable to upload Supplementary Table 1 to the system. We are very sorry for the inconvenience caused to you. Therefore, in this major revision, we have re-uploaded Supplementary Table 1 to the system. You will be able to find it in the system in the future. Finally, we would like to thank you again for your understanding of our negligence this time.
Thank you for your consideration.
- 2. Figure 2. Consensus maps built on fourteen original map datasets:
The authors need to put the actual chromosome length (Mbp) in the corresponding chromosome numbers.
Thanks for your positive comments. As suggested, we have revised Figure 2 and placed it in the supplementary materials. They are Figure S1 and Figure S2 respectively. In these two figures, we have clearly shown the actual length of the chromosomes. We then have re-submitted the manuscript.
Thank you for your consideration.
4.1. Literature Retrieval and QTL Data Collection of Maize Yield-Related Traits:
- The author fails to discuss how many plants they used in each type of generation, e.g., doubled haploid (DH) populations, F2:3 populations, F2 populations, recombinant inbred lines (RIL) populations etc. Please explain the details of experimental procedures here.
Thanks for your positive comments. As suggested, we made certain modifications to 4.1, and in Table 1 we have detailed statistics of the size of the groups involved in each study. We then have re-submitted the manuscript.
Thank you for your consideration.
- The author needs to check the reference lists of those that are cited in the text. All the titles and other contents should be unique to the journal format.
Thanks for your positive comments. As suggested, we carefully checked the format of all the references cited in the text and corrected the ones with incorrect formats. We then have re-submitted the manuscript.
Thank you for your consideration.
Reviewer 2
Comments and Suggestions for Authors
The article by Xin Li and co-authors concerns the identification of yield traits in Zea mays L. Maize is an extremely important commercial species, and in this context, the article is interesting and necessary The article is well planned. The scientific objective, “Identification of Yield Traits in Maize,” is well chosen and clearly expressed in the Introduction. It could be emphasized more in the Abstract (e.g., “our aim was...”).
Thanks for your positive comments. As suggested, we added the corresponding content were that “In this study, our aim was to identify the key genomic regions that regulate the formation of components in maize yield through bioinformatics methods.” in Lines13-15 on page 1 of the manuscript. We then re-submitted the manuscript.
Thank you for your consideration.
The authors skillfully presented the research problem against the backdrop of growing agricultural problems (climate change) and the work of other authors in the Introduction.
The most important results include: They identified 159 potential candidate genes that regulate the formation of maize yield within 43 MQTL intervals. These key genes are involved in a variety of metabolic processes, including signal transduction of four plant hormones, cell growth and development, sucrose-starch synthesis and metabolism, and reproductive reproduction.
The methods are well chosen. The conclusions are clear.
However, the discussion is a bit short. Could the authors refer to other species/genera? Are there similar studies? Authors in particular should refer to monocots.
Thanks for your positive comments. As suggested, wwe added the corresponding content were that “In addition, Dong et al. [12] conducted MQTL analysis and candidate gene identification using 765 original QTLS related to corn yield components collected from 56 independent studies, and reached a consensus on 65 related MQTLS and 5,203 candidate genes. And among the 23 MQTLS, 25 functional genes related to corn grain traits and the reported candidate genes were detected. It can be seen from this that MQTL analysis can be used to mine candidate genes related to the formation of maize yield.” in Lines 262-268 on page 9 of the manuscript. We then re-submitted the manuscript.
Thank you for your consideration.
Reviewer 3
Comments and Suggestions for Authors
In this study, 554 original QTLs related to 11 yield components of maize were collected. The consensus map was then constructed with a total length of 7,154.30 cM. Approximately 80.32% original QTLs were successfully projected on the consensus map, and unevenly distributed on the 10 chromosomes. 159 candidate genes were found in above all MQTLs intervals, of which, 29 genes encoding E3 ubiquitin protein ligase, that was related with kernel size and weight. Other genes were involved in multiple metabolic processes, including plant hormones signaling transduction, growth and development, sucrose-starch synthesis and metabolism, and reproductive growth. In general, the experiments were well-performed.
Thanks for your positive comments. As suggested, we modified the corresponding content. We then re-submitted the manuscript.
Thank you for your consideration.
(1) Not all abbreviations have been introduced with full names, for example GDB and CNKI in Abstract. By the way, the CNKI database may not be accessible to international users, and therefore should be excluded.
Thanks for your positive comments. As suggested, we replaced the "CNKI" database with the "Google Scholar" database. Furthermore, regarding the writing of the "MaizeGDB" database, after consulting a large number of literatures, we found that in all the literatures, its writing rules are like this. We then have re-submitted the manuscript.
Thank you for your consideration.
- The chromosomes in Figure 4 should NOT be marked with the green color. The white color or a light grey color should be used to show the gene distribution clearly.
Thanks for your positive comments. As suggested, we have attempted to change the color of the chromosomes to white or light gray, but the final effect is not as good as green. Therefore, we still want to use green as the color of the chromosomes. And for the candidate genes identified in different MQTL intervals on each chromosome, we have also distinguished them with different colors. We hope you can understand this. We hope you can understand. We then have re-submitted the manuscript.
Thank you for your consideration.
- Usages of MQTLs and the candidate genes in fieldcross-breeding should be further discussed.
Thanks for your positive comments. As suggested, we added the corresponding content were that “In this study, through MQTL analysis, we obtained multiple stable MQTL intervals and potential candidate genes that regulate the formation of corn yield, providing valuable information for high-yield breeding of corn. In subsequent research, based on the identified candidate genes, we can create mutants through gene editing technology for backcrossing verification in the field, and thereby cultivate corn varieties with high-yield performance.” in Lines 12-13 on page 1 the of manuscript. We then have re-submitted the manuscript.
Thank you for your consideration.
- In references, some journals' names are in a wrong format, such as [36] and [50]. All Latin names should be in italic, such as Zea mays in Ref. [52].
Thanks for your positive comments. As suggested, We have carefully examined the references involved in this article and made corrections to those with incorrect formats. Furthermore, regarding the format issue of the journal name in reference [36], the reason we wrote it this way is that it is a Chinese journal. The English name we found is this one, and we cannot find its abbreviation. Therefore, we are deeply sorry. We then have re-submitted the manuscript.
Thank you for your consideration.
Comments on the Quality of English Language
- The language should be polished with an English native speaker.
- For example in Abstract, "that were regulated kernel size and weight" --> that was related with kernel size and weight.
- cell growth and development --> plant growth and development
- reproductive reproduction --> reproductive growth
Thanks for your positive comments. As suggested, We have refined the language of the article to a certain extent and replaced the sentences proposed above. We then have re-submitted the manuscript.
Thank you for your consideration.
Best wishes!
Xiaoqiang Zhao Professor
State Key Laboratory of Aridland Crop Science, Gansu Agricultural University
- Email: zhaoxiaoq@gsau.edu.cn

Reviewer 2 Report
Comments and Suggestions for Authors
The article by Xin Li and co-authors concerns the identification of yield traits in Zea mays L. Maize is an extremely important commercial species, and in this context, the article is interesting and necessary. The article is well planned. The scientific objective, “Identification of Yield Traits in Maize,” is well chosen and clearly expressed in the Introduction. It could be emphasized more in the Abstract (e.g., “our aim was...”).
The authors skillfully presented the research problem against the backdrop of growing agricultural problems (climate change) and the work of other authors in the Introduction.
The most important results include: They identified 159 potential candidate genes that regulate the formation of maize yield within 43 MQTL intervals. These key genes are involved in a variety of metabolic processes, including signal transduction of four plant hormones, cell growth and development, sucrose-starch synthesis and metabolism, and reproductive reproduction.
The methods are well chosen. The conclusions are clear.
However, the discussion is a bit short. Could the authors refer to other species/genera? Are there similar studies? Authors in particular should refer to monocots.
Author Response

(The authors gave the same response as above.)

Reviewer 3 Report
Comments and Suggestions for Authors
In this study, 554 original QTLs related to 11 yield components of maize were collected. The consensus map was then constructed with a total length of 7,154.30 cM. Approximately 80.32% original QTLs were successfully projected on the consensus map, and unevenly distributed on the 10 chromosomes. 159 candidate genes were found in above all MQTLs intervals, of which, 29 genes encoding E3 ubiquitin protein ligase, that was related with kernel size and weight. Other genes were involved in multiple metabolic processes, including plant hormones signaling transduction, growth and development, sucrose-starch synthesis and metabolism, and reproductive growth. In general, the experiments were well-performed.
1) Not all abbreviations have been introduced with full names, for example GDB and CNKI in Abstract. By the way, the CNKI database may not be accessible to international users, and therefore should be excluded.
2) The chromosomes in Figure 4 should NOT be marked with the green color. The white color or a light grey color should be used to show the gene distribution clearly.
3) Usages of MQTLs and the candidate genes in field cross-breeding should be further discussed.
4) In references, some journals' names are in a wrong format, such as [36] and [50]. All Latin names should be in italic, such as Zea mays in Ref. [52].
Comments on the Quality of English LanguageThe language should be polished with an English native speaker.
For example in Abstract, "that were regulated kernel size and weight" --> that was related with kernel size and weight
cell growth and development --> plant growth and development
reproductive reproduction --> reproductive growth
Author Response

(The authors gave the same response as above.)
